# 5.8 GHz High-Efficiency RF–DC Converter Based on Common-Ground Multiple-Stack Structure

**DOI:** 10.3390/s19153257

**Published:** 2019-07-24

**Authors:** Jongseok Bae, Sang-Hwa Yi, Woojin Choi, Hyungmo Koo, Keum Cheol Hwang, Kang-Yoon Lee, Youngoo Yang

**Affiliations:** 1College of Information and Communication Engineering, Sungkyunkwan University, 2066 Seobur-ro, Jangan-gu, Suwon 16419, Korea; 2Electrical Environment Research Center, Korea Electrotechnology Research Institute, Changwon-si 51543, Korea

**Keywords:** microwave power transfer, rectenna array, rectifier, multiple-stack RF–DC converter, RF isolation network, voltage doubler

## Abstract

This paper presents a 5.8 GHz RF–DC converter for high conversion efficiency and high output voltage based on a common-ground and multiple–stack structure. An RF isolation network (RFIN) for the multiple-stack RF–DC converter is proposed to combine the DC output voltage of each stack without separating its RF ground from the DC ground. The RFIN is designed using micro-strip transmission lines on a single-layer printed circuit board (PCB) with a common ground for the bottom plate. A 4-stack RF–DC converter based on a class-F voltage doubler for each stack was implemented to verify the proposed RFIN for the multiple-stack and common-ground structure. The performances of the implemented 4-stack RF–DC converter were evaluated in comparison to the single-stack converter that was also implemented. The size of the implemented 4-stack RF–DC converter using bare-chip Schottky diodes is 24 mm × 123 mm on a single-layer PCB. For an input power of 21 dBm for each stack of the RF–DC converter with a load resistance of 4 kΩ, a high efficiency of 73.1% and a high DC output voltage of 34.2 V were obtained.

## 1. Introduction

Microwave power transmission (MPT) technology has been developed for various wireless charging applications, such as space solar power systems, internet of things (IoT) sensors, medical devices, RF energy harvesting systems, and mobile devices. The MPT system has the advantage of charging distance, compared to popular wireless charging systems that are based on electro-magnetic resonance or magnetic coupling methods [1,2,3,4,5,6,7,8]. However, the efficiency of the MPT system, which is mainly determined by the efficiency of the transmitter (Tx), path loss between the Tx and receiver (Rx), and the RF–DC conversion efficiency of the Rx, is very poor.

Figure 1 shows a block diagram of an MPT system based on massive Tx and Rx antenna arrays, which are required to reduce the path loss between the Tx and Rx. A phased array is required for the Tx to sharply form and precisely steer the antenna beam. The Rx usually consists of an antenna array, RF–DC converters or rectifiers, DC–DC converters, and the load [7]. One or multiple DC–DC converters are required between the load and the RF–DC converters to generate a constant and stable DC output voltage. The overall efficiency of the Rx is determined by the efficiencies of the RF–DC converter and the DC–DC converter.

If the required output DC voltage at the load is higher than the output voltage of the RF–DC converter, a boost DC–DC converter must be used. Otherwise, if the output voltage of the RF–DC converter is sufficiently higher than the required output DC voltage at the load, a buck DC–DC converter can be used, which is generally simpler and more efficient than a boost converter. For higher efficiency and simpler circuit for the Rx, it is often desirable for the RF–DC converter to have high output DC voltage, in addition to high efficiency.

The RF–DC converter consists of an input matching circuit, Schottky diodes, and a low-pass filter. The output DC voltage of the RF–DC converter is dependent on the circuit structure and the number of diode stack. Compact structures using a shunt diode or a series diode generally have the disadvantage of low output voltage. Since voltage doubler structures have two Schottky diodes, the output voltage becomes about twice as high as that of the single diode structures [5]. The output voltage can be boosted by increasing the number of diode stack, while sacrificing the conversion efficiency mainly due to the increased input matching loss.

Since RF–DC converters have been applied to various sensors that acquire energy or information for the operation from the input RF signal in addition to the MPT systems, a lot of previous studies of various aspects can be found [9,10,11,12,13,14,15,16,17]. Harmonic termination can be used to improve the conversion efficiency of the RF–DC converter based on the Schottky diode [9,10,11,12,13,14,15,16]. The authors in [9] realized a class-C operation at 2.45 GHz was realized using a shunt diode with short-circuited second and third harmonics, and reported an efficiency of 72.8% and an output voltage of 1.91 V using an input power of 8 dBm with a load resistance of 742 Ω. A Class-F input matching network with a short-circuited second harmonics and an open-circuited third harmonics was proposed [10,11]. The author in [10] reported a class-F rectifier at 2.45 GHz with a third harmonic matching circuit, and with even harmonic cancellation characteristics by using dual diodes and reported an efficiency of 75.4% at an input power of 20 dBm and a load resistance of 1 Ω. The authors in [11] reported an efficiency of 79.5% using a shunt diode at an input power of 17.65 dBm and a frequency of 5.8 GHz. The authors in [12] reported a class-F voltage doubler at 5.8 GHz with an efficiency of 71% and an output voltage of 5 V at an input power of 14.77 dBm and a load resistance of 1300 Ω.

For the massive Rx array, power-combining methods for the received power from each antenna are very important to increase the DC output power, and to generate the required DC voltage [18,19,20,21,22,23,24,25,26]. The RF power-combining methods can increase the input power to the RF–DC converter, though the combiner has considerable size and loss. The DC power-combining methods include current-combining and voltage-combining methods [18,19,20,21,22,23,24]. The authors in [26] introduced a hybrid power-combining rectenna array using both RF power-combining and DC power-combining methods for higher received power. However, power-combining methods have not yet been comprehensively analyzed and compared from the systems’ point of view yet.

In this paper, three power-combining methods of the Rx array for the MPT systems were briefly compared and analyzed. To increase the output DC voltage of the Rx for the MPT systems, an RFIN, which can be implemented using microstrip transmission lines on a single-layer PCB, is proposed for the common-ground and multiple-stack RF–DC converter. Compared to the conventional voltage-combining methods, the proposed voltage-combining circuit using the RFIN could be implemented on a single-layer PCB with a very simple structure. To verify the proposed RFIN, a 5.8 GHz 4-stack RF–DC converter using a class-F doubler as a unit stack was designed and implemented for high conversion efficiency and high output voltage. The measured performances of the implemented 4-stack RF–DC converter are presented in comparison to the implemented single-stack converter and some previously reported RF–DC converters.

## 2. Power-Combining Method

Figure 2 shows three representative power-combining methods for a rectenna array. For a massive Rx array, three methods can be selectively mixed and applied for the optimized system design. RF signals, which are received from multiple Rx antennas, can be directly combined using an RF power combiner for an RF–DC converter, as shown in Figure 2a. The input power to the RF–DC converter can be increased, but, in general, the RF power combiner has considerable loss. If the RF power received from each antenna is not enough for the RF–DC converter to have high efficiency or to work properly, the RF power-combining method must be considered.

Figure 2b,c are diagrams for the DC current-combining and the DC voltage-combining methods, respectively. Both methods can be categorized as DC power combining methods, which in general have relatively small loss and simple structure. For the current-combining Rx with N rectennas, the total output current ($I_{OUT}$) at the load is simply given as the sum of the output DC currents of the multiple parallel RF–DC converter stages:(1)$$I_{out}=I_{out,1}+I_{out,2}+\ldots +I_{out,N},$$
where $I_{out,N}$ is an output current of the N-th RF–DC converter. Then, the total power at the load ($P_{DC}$) and the output voltage ($V_{out}$ ) can be derived using $I_{out}$ as:(2)$$P_{DC}={\left(I_{out}\right)}^{2}\times R_{L},$$
(3)$$V_{out}=I_{out}\times R_{L}.$$

Figure 2c shows a voltage combining method using a multiple-stack RF–DC converter. The DC output of the *i*-th stack is fed to the RF ground of the (i+1)-th stack. Then, the output DC voltage of the N stack RC-DC converter ($V_{out}$) becomes about N times the output DC voltage of each stack.
(4)$$V_{out}=V_{out,1}+V_{out,2}+\ldots +V_{out,N},$$
where $V_{out,N}$ is the output voltage of the N-th RF–DC converter. Then, the total power at the load ($P_{DC}$) and the output current ($I_{out}$) can be derived using $V_{out}$ as:(5)$$P_{DC}=\frac{{\left(V_{out}\right)}^{2}}{R_{L}},$$
(6)$$I_{out}=\frac{V_{out}}{R_{L}}.$$

For the massive antenna array, the RF power-combining and two DC power combining methods can be appropriately used for the overall Rx system. The Rx structure for power-combining should be optimized through careful consideration of the nominal received power, the number of antennas, the required output DC voltage for the load, and so on.

## 3. Design of the Proposed RF–DC Converter

### 3.1. Common-Ground Multiple-Stack Structure

For the Rx with a massive antenna array, DC power-combining in either current or voltage form is required. In particular, appropriate use of the voltage-combining method allows the Rx to have optimal output DC voltage for a DC–DC converter with high efficiency. Figure 3a shows a schematic of the N-stack RF–DC converter with the conventional voltage-combining method. Since the DC output of the i-th stack, except the first stack, is fed to the RF ground of the (i+1)-th stack, the RF ground of the (i+1)-th stack must be separated from the DC ground. Though it is very easy and simple to combine the voltage outputs from the multiple rectennas, its implementation is not as simple as its concept. Since the bottom plate of a PCB is usually used as a ground for general RF circuits, each stack must be implemented on a separated PCB, or the whole circuit must be implemented on a multi-layer PCB.

Figure 3b is a schematic of the N-stack RF–DC converter with the proposed common-ground structure. Different from the conventional method, the RF and DC grounds for the proposed method are the same for all stacks. The RFIN provides the (i+1)-th stack with the DC output of the *i*-th stack, and also provides very low impedance at its input and output ports to isolate the RF signal. To use the whole bottom plate as common ground, the RFIN must be implemented on the top metal plate of the PCB. Section 3.3 presents how to design the RFIN.

### 3.2. Class-F Voltage Doubler

To design the overall multi-stack RF–DC converter using the proposed common-ground structure, a unit RF–DC converter stack should be designed. For high conversion efficiency, a class-F voltage doubler was designed for the 5.8 GHz band using a bare-chip Schottky diode, MACOM’s MA4E1319-1, which has a very low series resistance of 4.6 Ω, and a junction capacitance of 0.047 pF. Since the three-terminal chip has two Schottky diodes connected in series, it is more convenient to design a voltage doubler.

Figure 4 shows a source-pull setup for the class-F voltage doubler whose source impedance must be very low for the second harmonics, and very high for the third harmonics. Since the series connection of the two diodes looks perfectly balanced at the input for the second harmonics, no second harmonic voltage can be generated at the input [11]. Figure 5a shows the efficiency contours for the input third harmonics. Due to the internal parasitic capacitance, an optimum region can be found at the slightly inductive impedance area. $C_{L}$ at the output should be very large to have only DC voltage ($V_{out}$) at the output.

Figure 5b shows the simulated efficiencies according to the input power with and without third harmonic control. Using the fundamental source impedance of 180 + *j*32 Ω, the optimum third harmonic impedance of *j*150 Ω at an input power of 21 dBm, and load resistance of 1 kΩ, the class-F voltage doubler in the source-pull condition exhibited a high efficiency of 82.0% and output DC voltage of 9.6 V at input power of 21 dBm, while the normal voltage doubler without third harmonic control has a 5.2% lower maximum efficiency. Figure 5c shows voltage and current waveforms of the Schottky diodes.

Figure 6 shows the implemented input matching network for the fundamental and third harmonics of the proposed class-F voltage doubler. Figure 7 shows trajectories for the input impedance matching: (a) for third harmonic impedances and (b) for the fundamental impedances. The input matching network, including a bond-wire inductance of 0.3 nH, consists of series transmission lines and open stubs. In the matching network for the third harmonics, a 50 Ω open stub with an electrical length of 30${}^{\circ }$ is used to make a short circuit at point D for the third harmonics. Then, a 50 Ω series transmission line with an electrical length of 11${}^{\circ }$ and a bond-wire inductance rotate the impedance to the optimum point. The fundamental impedance matching network, before the 3rd harmonic matching network, consists of a 73${}^{\circ }$ series transmission line with a characteristic impedance of 50 Ω, a 65${}^{\circ }$ open stub with a characteristic impedance of 50 Ω, and a DC-blocking capacitor of $C_{DC}$. For accuracy, an electromagnetic field simulation tool, Keysight’s Momentum, was used in the design of the input matching network. The entire input matching network showed an insertion loss of 0.2 dB.

### 3.3. RFIN

Figure 8 shows (a) a schematic of the RFIN with its ideal configuration, and (b) its practical configuration including bond wires. The RFIN is a reciprocal network and is basically designed using transmission lines. The RFIN should provide a short circuit in DC through the ports, and a ground in RF for both ports. To make an RF ground, the impedances for the fundamental and harmonics at each port were minimized. In this design, higher-order harmonics, except the second, were ignored for circuit simplicity:(7)$$Z_{P1}\left(f_{0}\right)=Z_{P2}\left(f_{0}\right)=Z_{P1}\left(2f_{0}\right)=Z_{P2}\left(2f_{0}\right).$$
where $Z_{Pj}\left(f_{0}\right)$ and $Z_{Pj}\left(f_{0}\right)$ are the fundamental and second-harmonic impedances for the *j*-th port, respectively.

For the ideal configuration, the 2nd harmonic impedance of $Z_{1}\left(2f_{0}\right)$, which is the same as $Z_{P1}\left(2f_{0}\right)$, is obtained as 0 Ω using a 45${}^{\circ }$ open stub. In the same way, the fundamental impedance of $Z_{3}\left(f_{0}\right)$ becomes 0 using a 90${}^{\circ }$ open stub. Then, the fundamental impedance of $Z_{2}\left(f_{0}\right)$, which is the same as $Z_{P1}\left(f_{0}\right)$, becomes 0 using a 180${}^{\circ }$ series transmission line.

However, for the practical RFIN, bond wires, which connect the Schottky diode to the circuit board, should be considered. The bond wire can be modeled using an inductance, $L_{BOND}$. An inductance of about 0.3 nH was extracted for a bond wire using 3D electromagnetic (EM) simulation. The impedances for the fundamental and second harmonics of $Z_{P1^{\prime }}$ or $Z_{P2^{\prime }}$ are designed to reduce $Z_{P1}$ or $Z_{P2}$ to close to 0 Ω through $L_{BOND}$. An open stub, which was optimized to have an electrical length of 30${}^{\circ }$ and a characteristic impedance of 62 Ω, makes the second harmonic impedances of $Z_{P1}\left(2f_{0}\right)$ and $Z_{P2}\left(2f_{0}\right)$ almost 0 Ω, together with $L_{BOND}$. Additionally, the fundamental impedances of $Z_{P1}\left(f_{0}\right)$ and $Z_{P2}\left(f_{0}\right)$ were sent to 0 Ω using a 90${}^{\circ }$ open stub and an 170${}^{\circ }$ series transmission line both with a characteristic impedance of 62 Ω.

Figure 9 shows the simulation results for the fundamental and second harmonic impedances of $Z_{P1}$ (after the bond wire) and $Z_{P1^{\prime }}$ (before the bond wire) on the Smith chart. The fundamental and second harmonic impedances after the bond wire were obtained using an EM simulation as 0.65−j2.1 and 0.24+j1.9 Ω, respectively. The simulated impedances are very close to 0 Ω, which is very important for the performances of the overall multi-stack RF–DC converter.

### 3.4. Experiomental Verification

Figure 10 shows a schematic of the implemented 4-stack RF–DC converter based on class-F voltage doublers. The 4-stack RF–DC converter was designed on a single dielectric layer PCB with a bottom plate as a common ground for both RF and DC. The PCB for circuit implementation is a Taconic’s RF-35 with a dielectric thickness of 20 mil, a relative permittivity of 3.5, and a loss tangent of 0.0018. The proposed RFIN isolates the RF signal, and feeds the DC voltage from one stack to the next stack. The same RF power must be supplied to each voltage doubler stack. $C_{L}$ of 3 pF was used at the output.

Figure 11 shows a photograph of the implemented 4-stack RF–DC converter based on class-F voltage doublers. The size of the implemented circuit is 24 mm × 123 mm. The Schottky diode chip, MA4E1319-1, was mounted as chip-on-board (COB). ATC’s high-Q capacitors were used for both the input matching and the load. Figure 12 shows a photograph of the measurement setup. After a drive amplifier, a 4-way Wilkinson power divider equally splits the input power and supplies to the input of each voltage doubler stack. An electrical load, Maynuo’s M9712, is used at the load to provide optimized load resistance. The efficiency of the 4-stack RF–DC converter was calculated using the measured DC voltage and current.

Figure 13 shows the simulated and measured efficiencies and output DC voltages for the implemented single-stack and 4-stack RF–DC converters. The input RF signal is a continuous-wave (CW) with a frequency of 5.8 GHz. The single-stack RF–DC converter demonstrated an efficiency of 73.6% and an output DC voltage of 8.58 V at an RL of 1 kΩ, and an input power of 21 dBm. The 4-stack RF–DC converter demonstrated an efficiency of 73.1% and an output DC voltage of 34.2 V at an RL of 4 kΩ, and an input power per each stack of 21 dBm. The 4-stack has almost the same efficiency and four times higher output voltage compared to the single-stack. Figure 14 shows the simulated and measured efficiencies (in a) and output DC voltages (in b) of the implemented 4-stack RF–DC converter according to the various load resistances (2, 4, 6, 8, and 10 kΩ). Since the rectifier was designed to have an optimum load of 4 kΩ, it exhibited somewhat degraded efficiencies with other load resistances. However, as the load increases, the output DC voltage increases. Table 1 shows the performance comparison to the previously reported RF–DC converters using Schottky diodes at the frequency range (5–6) GHz. Compared to the previous works, the proposed 4-stack RF–DC converter exhibited very high output DC voltage and high efficiency.

## 4. Conclusions

In this paper, a 5.8 GHz RF–DC converter with a common-ground and multiple-stack structure using the RFIN was presented for high efficiency and high output DC voltage. The proposed RFIN was designed to isolate the RF signal, and to feed the DC voltage from one stack to the next stack, so that the DC voltage of each stack is combined at the output. The RFIN is composed of open-stubs, series transmission lines, and bond wires to have short-circuited fundamental and second harmonics for both ports. Since the RFIN is based on micro-strip transmission lines, the multi-stack RF–DC converter can be implemented on a PCB with a single dielectric layer for its bottom plate as a common ground for both RF and DC.

A 5.8 GHz class-F voltage doubler was designed using bare-chip Schottky diodes for a unit stack of the RF–DC converter to obtain high efficiency. It has an optimum termination circuit for the third harmonics as well as the fundamental matching network at the input. Based on the class-F voltage doubler and RFIN, a single-stack and 4-stack RF–DC converters were implemented for comparison and verification.

The implemented 4-stack RF–DC converter was evaluated by exciting a CW signal with a frequency of 5.8 GHz. With an input power of 21 dBm for each input port, an efficiency of 73.1% and an output DC voltage of as high as 34.2 V were achieved at an RL of 4 kΩ, while the implemented single-stack RF–DC converter exhibited almost similar efficiency and four times lower output voltage. The proposed multiple-stack RF–DC converter can be applied to the Rx’s of the MPT systems with massive antenna array. It can be used even in conjunction with other power-combining methods, such as RF power-combining, current-combining, or conventional voltage-combining methods for optimum system design.

## Figures and Tables

**Figure 1 sensors-19-03257-f001:**
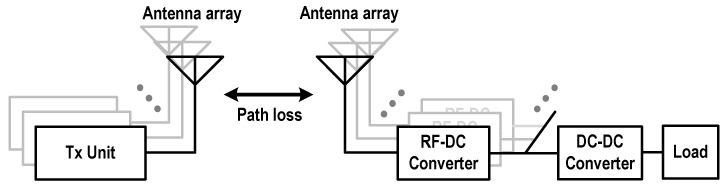
A block diagram of the MPT system based on massive Tx and Rx antenna arrays.

**Figure 2 sensors-19-03257-f002:**
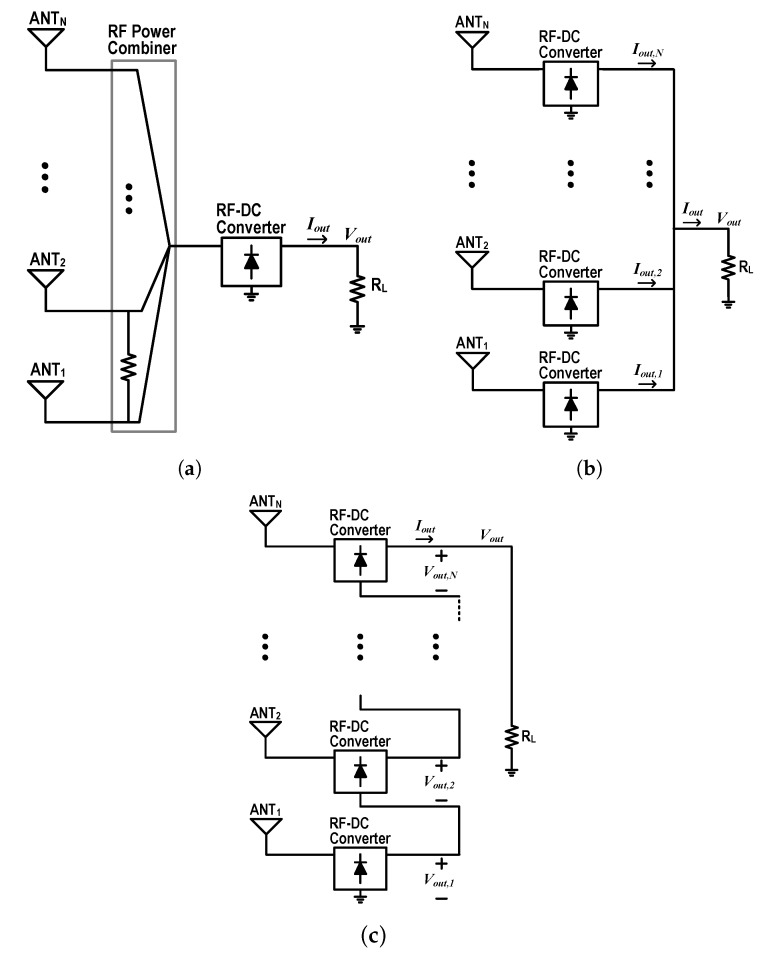
Power-combining methods for rectenna array, (**a**) RF power-combining; (**b**) DC current-combining; and (**c**) DC voltage-combining.

**Figure 3 sensors-19-03257-f003:**
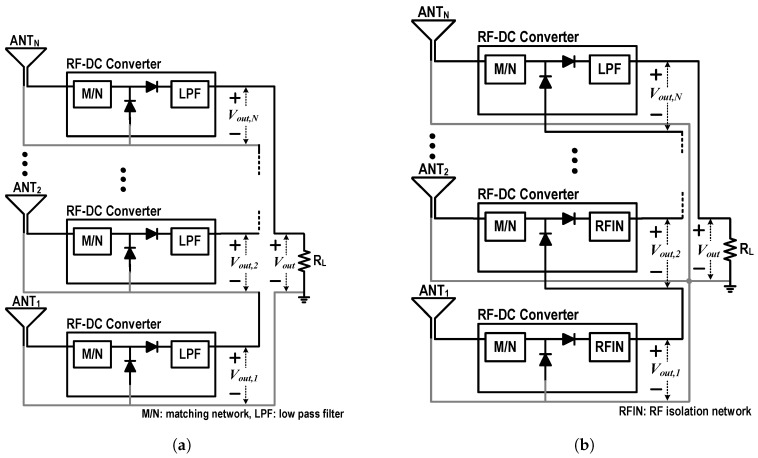
Schematics of the voltage combined N-stack RF–DC converters, (**a**) conventional and (**b**) proposed.

**Figure 4 sensors-19-03257-f004:**
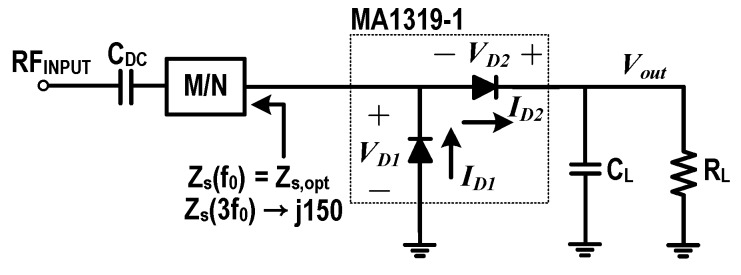
A source-pull setup for the class-F voltage doubler design.

**Figure 5 sensors-19-03257-f005:**
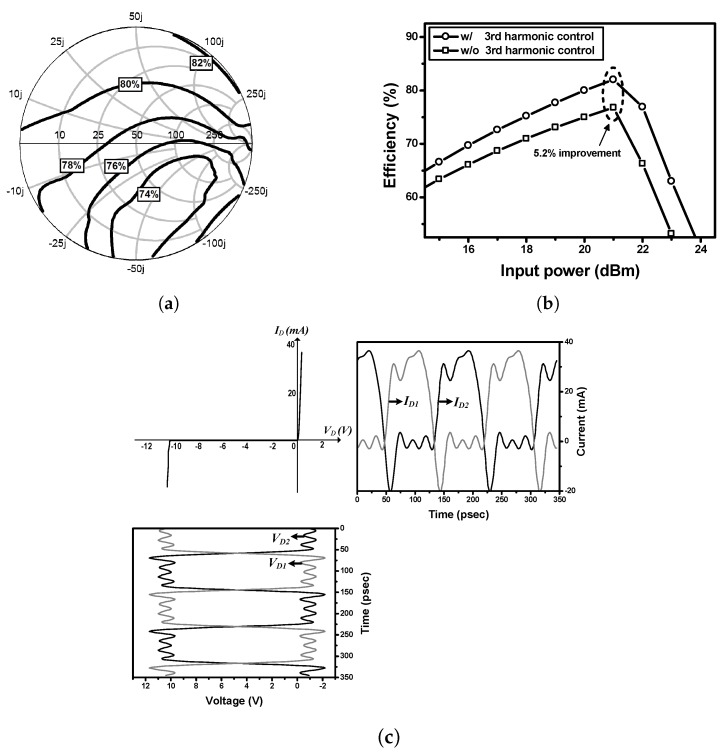
Simulation results of the class-F voltage doubler, (**a**) efficiency contours for the input third harmonics; (**b**) simulated efficiencies; and (**c**) voltage and current waveforms of the Schottky diodes.

**Figure 6 sensors-19-03257-f006:**
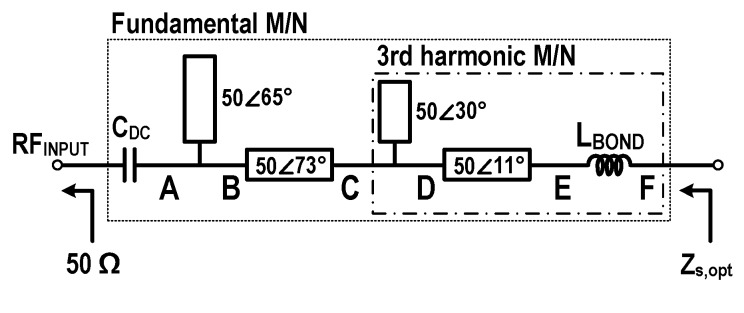
Schematic of the input matching network.

**Figure 7 sensors-19-03257-f007:**
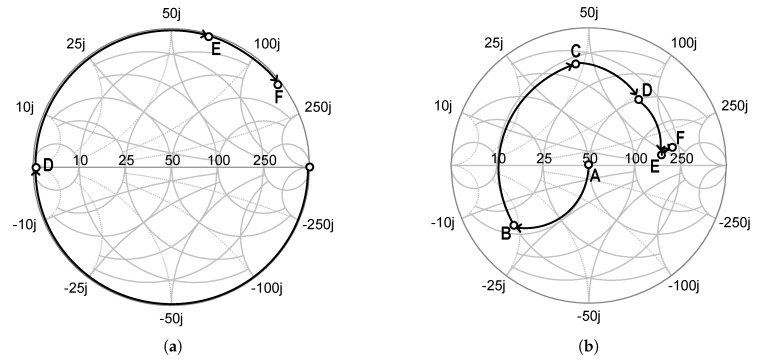
Trajectories for input impedance matching, (**a**) third harmonics and (**b**) fundamental.

**Figure 8 sensors-19-03257-f008:**
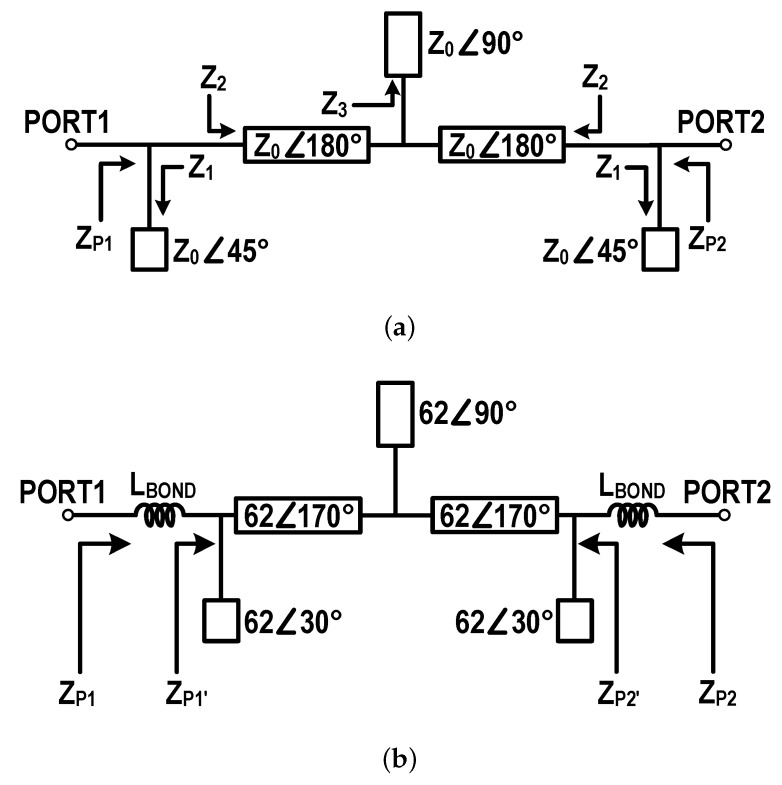
Schematics of the RFIN, (**a**) ideal configuration and (**b**) practical configuration including bond wires.

**Figure 9 sensors-19-03257-f009:**
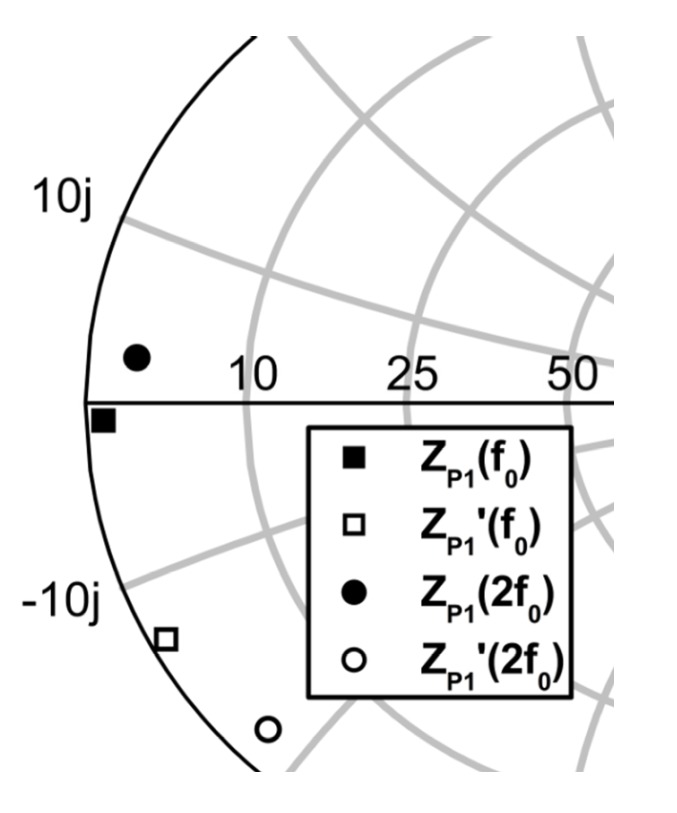
Fundamental and second harmonic impedances before and after the bond wire.

**Figure 10 sensors-19-03257-f010:**
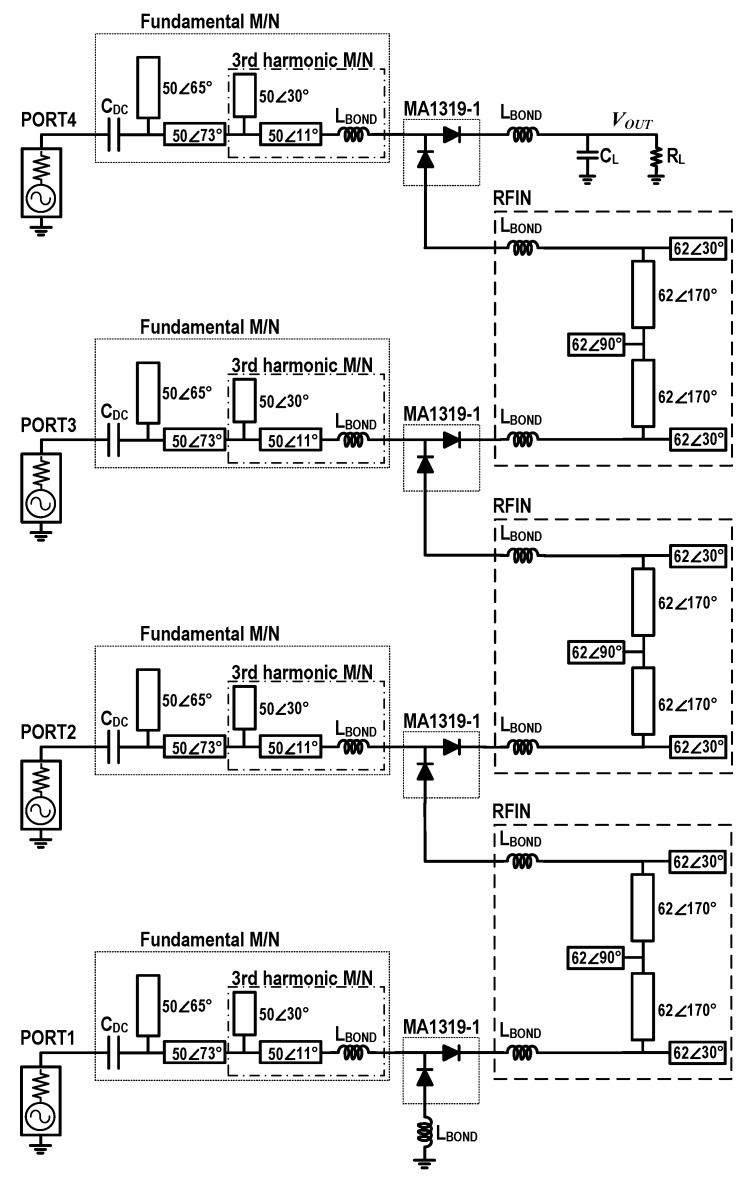
A schematic of the implemented 4-stack RF–DC converter based on class-F voltage doublers.

**Figure 11 sensors-19-03257-f011:**
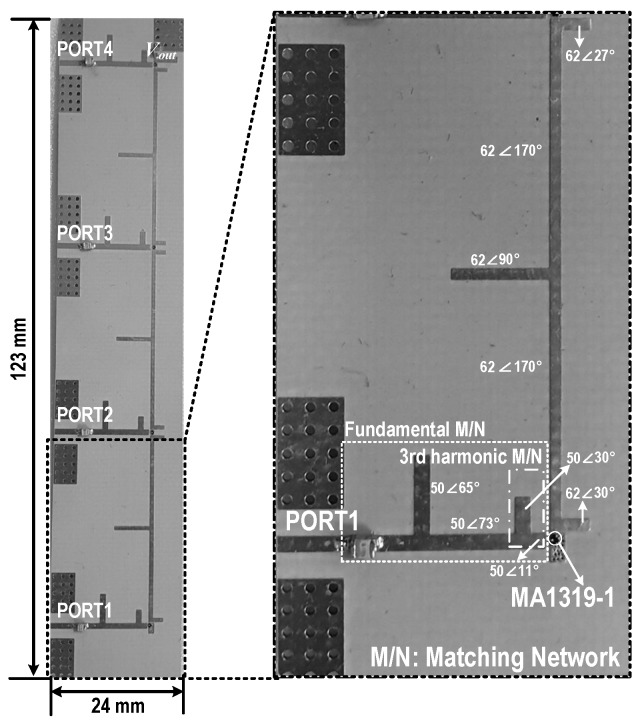
A photograph of the implemented 4-stack RF–DC converter.

**Figure 12 sensors-19-03257-f012:**
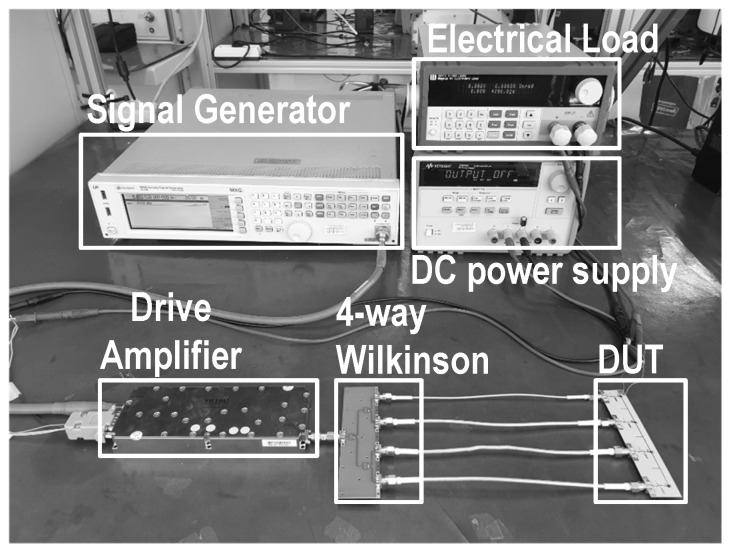
A photograph of the measurement setup.

**Figure 13 sensors-19-03257-f013:**
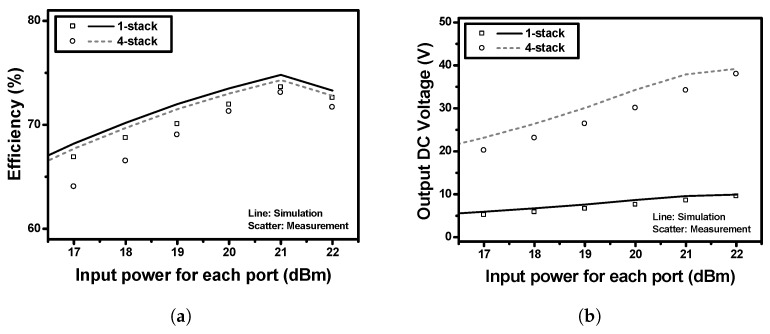
Simulated and measured results of the implemented single-stack and 4-stack RF–DC converters, (**a**) efficiencies and (**b**) output DC voltages.

**Figure 14 sensors-19-03257-f014:**
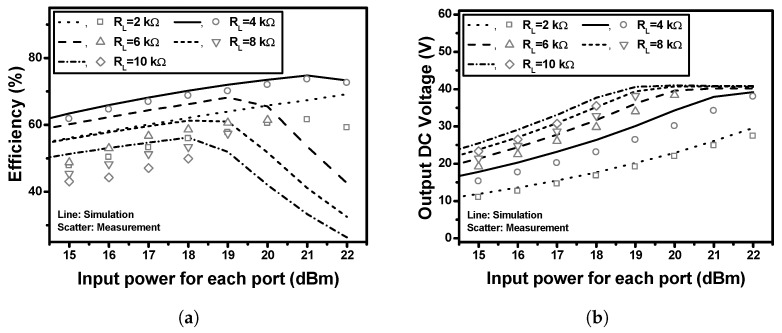
Simulated and measured results of the implemented 4-stack RF–DC converter according to the various load resistances, (**a**) efficiencies and (**b**) output DC voltages.

**Table 1 sensors-19-03257-t001:** Performance comparison to the previous works.

Ref.	Freq. (GHz)	Circuit Topology	Device Type	P_IN_ (dBm)	$R_{L}$ (Ω)	Output Voltage (V)	Peak Eff. (%)	PCB Size ($\lambda_{0}^{2}$)	Technique
[11]	5.8	Shunt diode	MA4E1317	17.65	N/A	N/A	79.5	N/A	Harmonic control (Class-F)
[12]	5.8	Voltage doubler	HSMS2860	14.77	1300	5.2 *	71.0	N/A	Harmonic control (Class-F)
[13]	5.8	Series diode	HSMS2860	10	N/A	2.3	51.5	0.67 × 0.50	Dual band matching
[14]	5.8	Shunt diode	HSMS286B	9	1300	N/A	60.6	1.12 × 0.48 *	Optimizaion of output ripple
[15]	5.2	Voltage doubler	HSMS286C	24	1150	5.1	64.1	0.42 × 0.33	Harmonic control (Class-F)
This work	5.8	Single-stack Voltage doubler	MA4E1319-1	21	1000	8.58	73.6	0.46 × 0.37	Harmonic control (Class-F)
		4-stack Voltage doubler		27	4000	34.2	73.1	0.46 × 2.38	RFIN

* graphically estimated; $\lambda_{0}$: wavelength at center frequency.

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
