# Peer review of "5.8 GHz High-Efficiency RF–DC Converter Based on Common-Ground Multiple-Stack Structure"

_sensors, 2019, doi:10.3390/s19153257_

Round 1
Reviewer 1 Report
This
paper presents a rectifier array works at 5.8 GHz for microwave power
transmission with high input power. The theoretical analysis and
experimental validations have been properly presented. However, I have
the following concerns on this paper which need further justifications. 1. First
of all, the selling-point of this paper, as highlighted by the authors,
is the common ground arrangement and RF isolation between each
rectifier. But, the idea is not new since that the shared ground plane
on a single PCB board for rectenna arrays has been heavily investigated.
On the other hand, the so-called RF isolation network is also not new
and somehow complicated. A simpler method can be realized by using a
series inductor. [Ref
1] F. Erkmen, T. S. Almoneef and O. M. Ramahi, "Scalable
Electromagnetic Energy Harvesting Using Frequency-Selective Surfaces,"
in IEEE Transactions on Microwave Theory and Techniques, vol. 66, no. 5,
pp. 2433-2441, May 2018. [Ref
2] T. S. Almoneef, F. Erkmen, M. A. Alotaibi and O. M. Ramahi, "A New
Approach to Microwave Rectennas Using Tightly Coupled Antennas," in IEEE
Transactions on Antennas and Propagation, vol. 66, no. 4, pp.
1714-1724, April 2018. 2. In
Fig. 5 (b), the authors have shown the difference of efficiency when
the third harmonic termination stub is used/unused. In the text, the
underlying principle of the difference should be studied rather than
using simple sentences to summarize the phenomenon. 3. When
discussing the input source impedance of the rectifier, please note
that such impedance is power-dependent and also load-dependent. The
authors should specify the corresponding power and load resistance for
all mentioned impedance values. 4. From
Fig. 6, the DC-blocking capacitor Cdc is normally placed after the
matching network. In addition, this capacitor should be included to the
voltage doubler schematic as in Fig. 4. 5. Please study the difference when the Cdc is located in front/behind of the matching circuits. 6. Please replace Fig. 5c by using I-V diagram of the diode junction which is more straightforward. 7. The
effect on performance with and without using the RFIN should be shown.
As the current structure in the design is frequency-dependent and a bit
complicated. 8. The size of the circuit as shown in Fig. 11 is large. Please compare the electrical size of all circuits in Table I. 9. When
the rectifier arrays are in series connections, the step rectifiers
will have a significant impact on the load impedance of the former
rectifier. This work also has alternative RFIN between rectifiers which
influence the load as well. Do you have any comments? 10. Please show the results of conversion efficiency vs. load. The recommended range is from 1 to 10k ohms.
Reviewer 2 Report
Please refer the attachment.

Reviewer 3 Report
This is a good paper, a 5.8GHz RF-DC converter is proposed and designed. The written of this paper is very good. The system structure is provided, and the detailed circuit is also provided. At last the PCB prototype is presented with the testing platform. The experimental results validate the proposed structure. Here are a few minor suggestions.
1. Please clarify the contributions of the proposed system at the very beginning. For example, summarize in three bullets. It will help the reader to get the main points faster.
2. More experiments can be provided. In this submission, there are only two figures provided in Fig. 13. It is expected to have more experimental results to support this work.
Round 2
Reviewer 1 Report
Thanks for the response to my comments. Most of them have been addressed. One minor comment is that the dimension in the comparison table should be the electrical size (relative wavelength) since other designs work at different frequencies.
The authors stated that the circuit size is acceptable after taking the antenna part into account. This is not very accurate since some antennas like dipole and monopole must be detached from the metal ground.
